# How to Teach Photosynthesis? A Review of Academic Research

**Kateřina Jančaříková** [1,2,*] and **Antonín Jančařík** [1]

1   Faculty of Education, Charles University, 110 00 Stare Mesto, Czech Republic
2   Faculty of Science, Jan Evangelista Purkyne University, 400 96 Usti nad Labem, Czech Republic
*   Correspondence: katerina.jancarikova@pedf.cuni.cz; Tel.: +420-221-900-175

**Abstract:** This review study focuses on the teaching of photosynthesis, and it builds on previous review studies. The aim of this study is to present the findings to readers and thus contribute to improving the training of future teachers and student education. This study conducted a critical review of the topic of photosynthesis education, by examining 80 systematically selected articles through quantitative and qualitative approaches. The quantitative analysis highlighted the increasing number of studies in recent years and helped identify the main issues, namely teaching methods and misconceptions. In contrast, the qualitative conceptualization showed that photosynthesis education is a key topic for the development of science literacy and serves as a model for how teachers can educate students on difficult and complex issues. An alarming issue is that misconceptions adopted in childhood are powerful and often persist into adulthood (including teachers). Professional education on photosynthesis is not possible without pedagogical content knowledge, the identification of student concepts, activating methods of learning, and methodical work with scientific language and thinking.

**Keywords:** misconceptions; science education; photosynthesis; biology education; literature review

## 1. Introduction

Photosynthesis is one of the most important metabolic processes and has long been a core part of the school biology curriculum in all countries, especially at secondary level. Photosynthesis and respiration play an essential role in the scientific understanding of biology, because it makes possible understanding the energy flows in organisms and ecosystems. Such awareness is critical, both for the acquisition of basic scientific knowledge, consisting in the awareness of the interconnection of individual components of living nature and their irreplaceability, and thus the formation of desirable environmental attitudes and responsible behavior based on them [1–3].

Czech students of biology education (future teachers in primary and secondary schools), whom we interacted with over the past ten years, have often had difficulty understanding photosynthesis, and then the broader ecological context (including impacts on food chains, food supply, and climate change) because of the lack of understanding of photosynthesis. Problems with their own understanding of the content are also transferred to the way they teach photosynthesis. This has negative effects on their future pupils. We started looking for ways to support them. Therefore, we focused on this topic.

We searched for review studies on the topic of learning photosynthesis [3–5].

Cañal [5] pointed out that students have difficulty understanding photosynthesis across countries, and misconceptions can begin forming at an early age. He cites several examples, including the inverse respiration misconception [6–8].

Barker and Carr [4,9] and Métioui et al. [3] focused on strategies and approaches to teaching photosynthesis, and made suggestions for innovative techniques to predict misconceptions. Barker and Carr [4,9] identified three existing strategies (the guided discovery strategy, the element analysis strategy, and the meaning of plant food strategy) and after pointing out their weakness, they developed a "generative learning strategy"

based on the "generative learning model". Métioui et al. [3] recommend introducing history into science teaching and use of a "history of the science approach".

In the six years that have passed since the last survey, many articles on teaching photosynthesis have been written. We therefore decided to do another review study.

We conducted systematic research to answer the question, "How to teach photosynthesis?" The aim of this study was to perform a quantitative and qualitative analysis of research on the topic of photosynthesis education and thus contribute to improving the training of future teachers and student education. In this paper, we will mainly present the conclusions that relate to misconceptions in the understanding of photosynthesis.

## 2. Materials and Methods

### 2.1. Characteristics of the Sample

We conducted systematic research of journal articles registered in the Web of ScienceTM (WoS, https://www.webofscience.com/wos, accessed on 1 September 2022). We used the following procedure to find articles included in the research. First, we limited the articles by topic, using keywords. We used "photosynthesis" as a search keyword in the topic. As we were only interested in articles on education, we limited the area of research in the second step to "education, educational research" and "scientific education disciplines". We identified 244 articles published in peer-reviewed journals and studying the teaching of photosynthesis until 20 March 2020. The number of published studies has grown exponentially since 1980 and doubled approximately once every 10 years (see cumulative graph in the upper left corner of Figure 1). Half the studies were published after 2010 (see Figure 1).

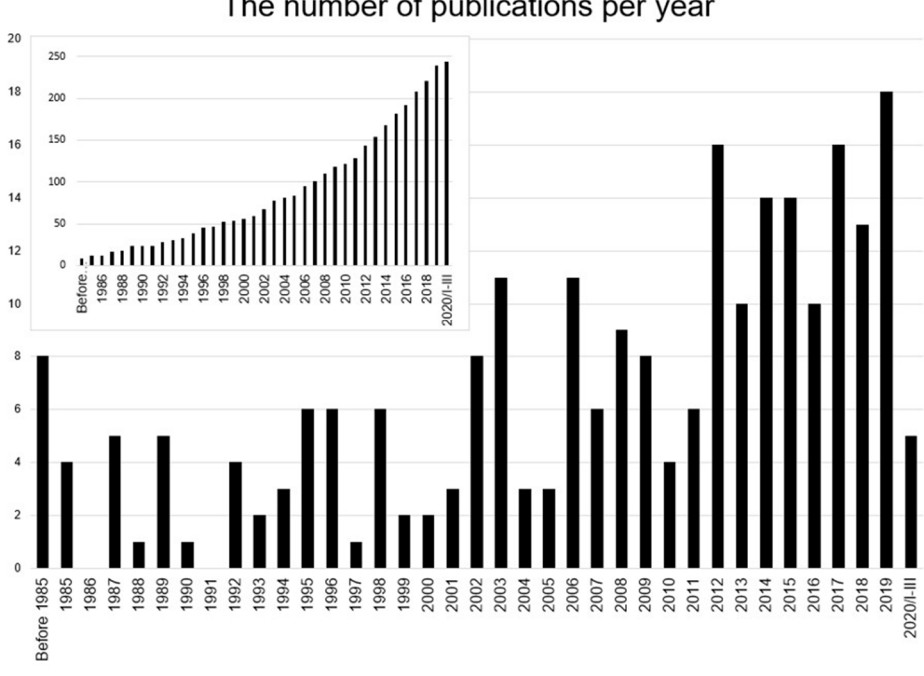

**Figure 1.** The number publications on "photosynthesis education" in WoS per year (Σ 244), in the upper left corner is a cumulative graph.

### 2.2. Selection of Articles

Following the procedures established in the PRISMA 2019 statement [10], selection was performed to obtain articles for the analysis (see Figure 2). We aimed to analyze only the articles to which the professional community pays attention, our criterion was that the article should have been cited five times or more. Based on this, 67 articles were discarded.

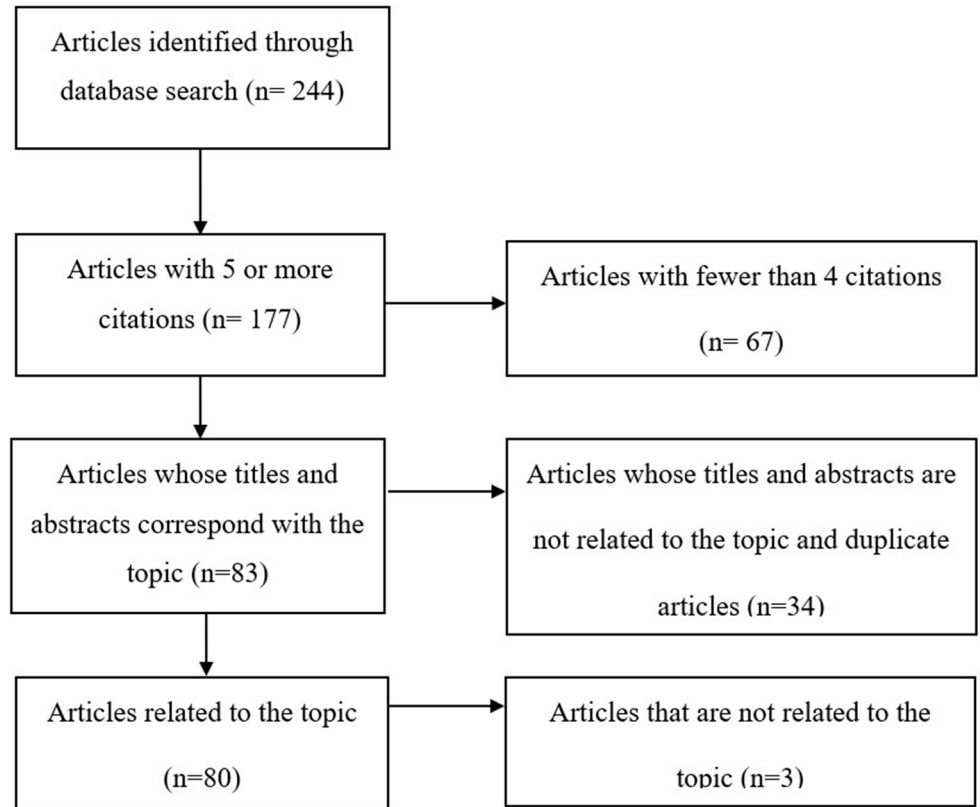

**Figure 2.** PRISMA flow diagram: the selection of 80 articles about photosynthesis education.

We looked at the titles and abstracts of the remaining 117 articles, to see if their topics were related to our study, and discarded 34 of them. Next, we read the remaining 83 articles, of which three did not correspond to the purpose of the study. Finally, 80 articles (75 research papers, 4 reviews, and 1 methodology) from 22 different journals were selected for analysis. The qualitative analysis was performed following the grounded theory of Strauss and Corbin [11]. In the qualitative analysis, we will focus on the most frequent concepts; i.e., misconceptions.

## 3. Results

### 3.1. The Quantitative Analysis

Most of the selected articles were from International Journal of Science Education (17), Journal of Biological Education (14), and Journal of Research in Science Teaching (12) (see Table 1).

**Table 1.** Journals with more than five published articles.

| Name of Journal | Total | Selected | Quartile |
|---|---|---|---|
| Journal of Biological Education | 43 | 14 | Q4 |
| American Biology Teacher | 32 | 8 | Q4 |
| International Journal of Science Education | 24 | 17 | Q3 |
| Journal of Chemical Education | 17 | 1 | Q3 |
| Journal of Baltic Science Education | 14 | 0 | Q4 |
| Journal of Research in Science Teaching | 13 | 12 | Q1 |
| Biochemistry and Molecular Biology Education | 12 | 1 | Q4 |
| CBE Life Sciences Education | 11 | 5 | Q2 |
| Research in Science Education | 7 | 4 | Q2 |

The number of research papers with only qualitative or only quantitative approaches were more or less equal (25 and 27 articles, respectively). The number of studies that used

combined research methods (20 articles) has increased over the last 10 years (Figure 3). We also identified five articles that could not be included in any of these categories.

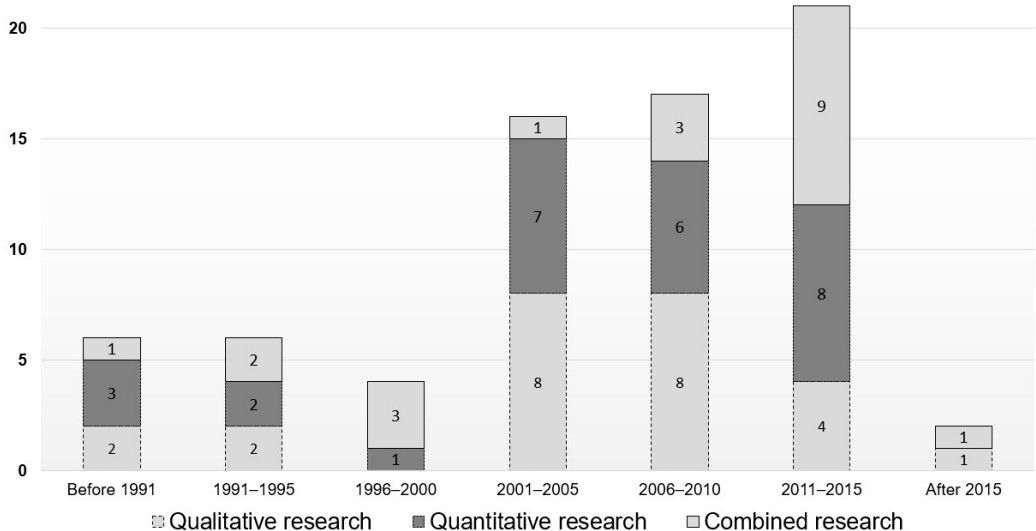

**Figure 3.** Number of publications and types of research methods over the years.

We also found out who the respondents were. We divided the articles according to the age group of the respondents (see Figure 4—university represents university students and adult biology teachers). Research that focused on multiple age groups was included in multiple categories.

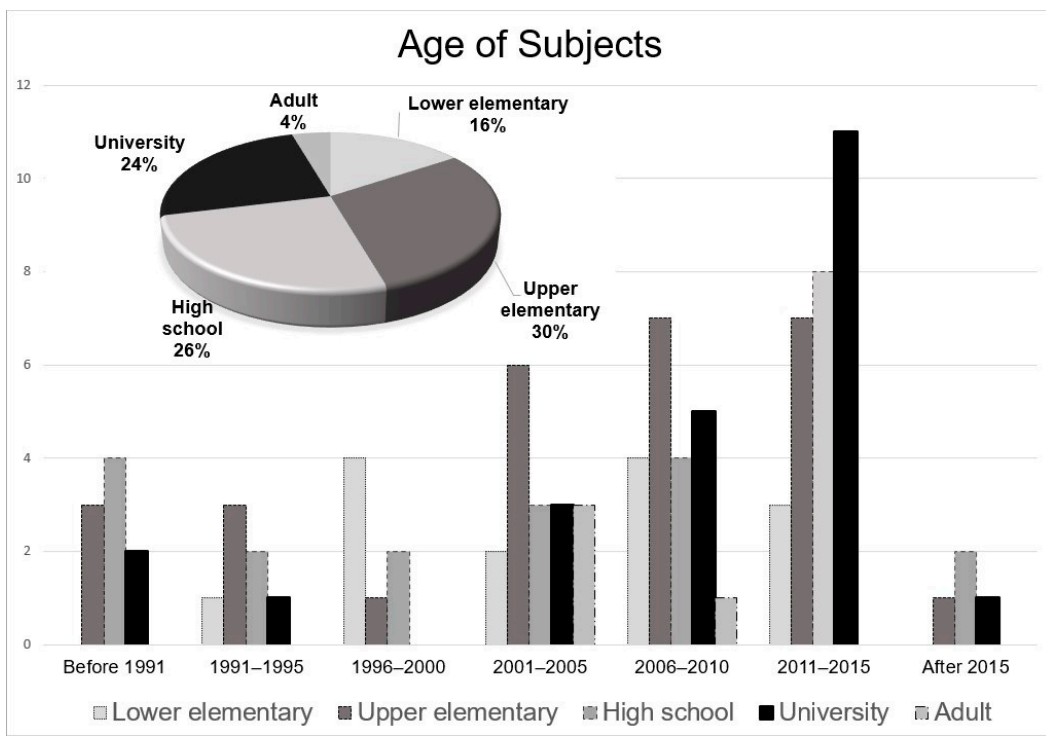

**Figure 4.** Level of Education of the Respondents.

The analyzed research took place in various countries globally, with the USA (29) and Turkey (9) ranking highest. Only one of the monitored studies came from a post-communist country (Czech Republic).

The analysis of the texts revealed three main topics of didactic research in the field of education of photosynthesis, namely

1.  monitoring misconceptions in understanding the content,
2.  methods of teaching (including using of ICT),
3.  analyzing of textbooks and curriculums.

The use of information and communication technologies (ICT) has become one of the main, but not the dominant, topics since 2000. The last two topics (the analysis of textbooks and curriculums) were identified but not addressed as often in the research (Figure 5).

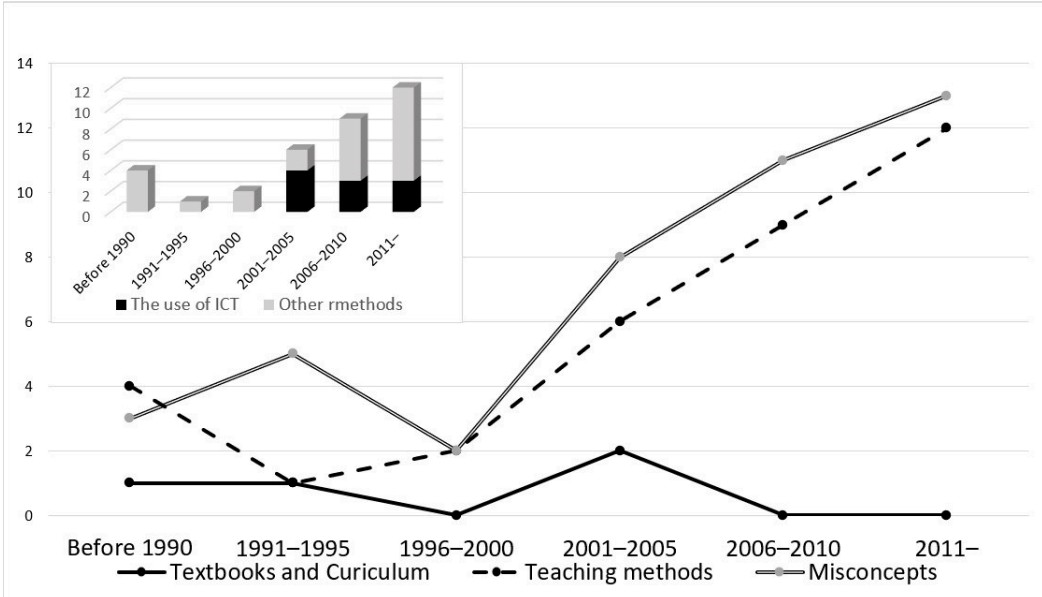

**Figure 5.** The main topics of the articles over the years and interest in the use of ICT as a method.

### 3.2. The Qualitative Analysis

The authors of the analyzed articles presented photosynthesis as a fundamental and crucial topic in science and environmental education. Understanding it, and the broader context of the carbon cycle and energy concepts, were considered crucial to the promotion of ecology ([12] as cited in [13]) and scientific (biological) literacy among students and citizens at large, in addition to environmental science literacy ([14–18], and also [19–22] as cited in [18]).

Photosynthesis is marked one of the most challenging topics in education, because it requires an interdisciplinary approach [23,24]. Knowledge of chemistry, biology, physics, physiology, and ecology is necessary in order to understand photosynthesis [8]. Waheed and Lucas [24] found that very few (5 out of 56) students (aged 14 years) demonstrated an understanding of all the ecological, biochemical, physiological, and energy change aspects of photosynthesis. Most (93%) students showed an understanding of the ecological aspect. A few (20%) showed that they understood the energy change aspect. Most showed an understanding of the physiological and biochemical aspects (57% and 68%, respectively).

Photosynthesis is characterized by a number of conceptual aspects and concepts [25]. For example, cellular respiration, has proven to be very difficult for learners, both in high school and at the university [26].

All research dealing with the understanding of photosynthesis points to a large amount of poor understandings or misunderstandings in the target populations, ranging from the youngest pupils to adults (university students and teachers). Misunderstandings often lead to misconceptions or alternative conceptions. Misconceptions are in opposition to the currently accepted scientific views [13] (p. 72). Constructivist researchers often examine misconceptions. Finding out what students are thinking and uncovering their misconceptions is considered the fundamental starting points in helping students overcome

the barriers to understanding, as uncovered by the research mentioned above [27]. Düsing et al. [28] pointed out that German pupils (aged 13–16 years) did not adequately understand photosynthesis and the carbon cycle of fossil fuels. They stated that, while describing ecosystems, students tended to omit decomposers; that is, they neither perceived the cycling of matter nor understand the importance of mineralization. Most of them mentioned only one compound with carbon and carbon dioxide. The presence of carbon was reported by most of them only in living bodies. Pupils did not realize that carbon is a fossil fuel and did not understand its role in industry, transportation, etc. They did not understand the environmental problems caused by anthropogenic impacts [29]. Similar results emerged from different countries, with different histories and curriculums [5].

Although older respondents are believed to make fewer mistakes, similar misconceptions were found among future teachers (students of teaching) as those in primary school pupils [1,30]. Eisen and Stavy [29] compared two groups of Israel university students (62 students of biology and 126 students of non-biology majors).

Their results indicated that people who did not study advanced courses in biology showed very little understanding of the essential role of photosynthesis in the ecosystem. They concluded that teaching of biology in high schools is not effective. Many authors pointed out that students do not understand the function of the ecosystem as an interrelated whole. That they do not realize that photosynthesis, respiration, and decay are not related to a view of the cycling of matter in ecosystems [20].

Käpylä et al. [31] found out that only 3 out of 10 of students—future primary teachers involved in this study—fully understood photosynthesis. They had a scientific understanding of the process, as they described all substances necessary for photosynthesis to produce organic substances, sugar, starch, and other carbohydrates. Some students, and future teachers, had conceptual shortages and/or misconceptions, as seen in their descriptions of photosynthesis. The following types of problems were discovered:

- four students did not mention carbon dioxide as a necessary starting material,
- three did not mention sugar as the reaction product,
- three forgot that oxygen was released in the reaction,
- two students thought that carbon dioxide was released in the process.

As expected, students who were future biology teachers had a better scientific understanding of photosynthesis. However, even in this group of future teachers, the scientific understanding was not without problems. Two (out of 10) did not mention that oxygen was released during the process [31].

Hazel and Prosser [32] focused on understanding of photosynthesis of university students from Australia. They used concept maps to elicit student's views of photosynthesis before and after studying photosynthesis in a general biology course. Although students achieved better results after the course, their progress was not as high as expected. Their misconceptions around photosynthesis were more deeply rooted than the lecturers expected. Therefore, it is a mistake to assume that university students understand photosynthesis enough for a university educator to explain its molecular details. Before the professional explanation, it is necessary to determine whether the students really have the basic knowledge that can be augmented.

All authors based their proposals for teaching innovation on constructivism, which was applied consistently. This includes working with preconceptions, individualization, action and experiential teaching, and self-reflection, accompanied by the social dimension of education. The authors of the analyzed articles perceived many problems accompanying the teaching of photosynthesis. They described difficulties in understanding photosynthesis and misconceptions. Similar results were obtained from different countries with different education systems. Ways to improve the teaching of photosynthesis have been suggested in several articles, with recommendations for innovative approaches and activating strategies (including the use of ICT or the creation of analogies) and support for the teachers. Despite this, only five studies focused on teachers. These studies showed that some teachers did not sufficiently understand photosynthesis. Misconceptions were discovered among teachers

and students of pedagogical disciplines (future teachers). The misunderstanding of teachers is one of the reasons why teaching fails.

Misunderstanding photosynthesis is a risk to society. People who do not understand it do not even understand environmental problems and will mishandle them. For example Magntorn and Helldén [33] conducted qualitative research of 15 secondary school pupils (aged 13–14 years) and found that none of the students made reference to fossil fuels, and surprisingly few students acknowledged cars and industry. The students were unaware of the fundamental role of decomposers and the release of carbon dioxide into the atmosphere through cellular respiration [13]. Analysis of the components that students considered relevant for the carbon cycle and how they connected them revealed that the students lacked an awareness of both the anthropogenic impact and the full range of functional groups involved in the cycling of matter [15,34,35].

Students require higher cognitive thinking and a high level of scientific abstraction in order to understand such complex topics as photosynthesis, with many conceptual aspects such as biological, ecological, physiological, biochemical, molecular biological, and physical ones. Käpylä et al. (2009) pointed that the more respondents know about a topic, the more difficult it is for them. Specifically, in their study, none of the 10 first-grade teachers considered the conceptual difficulties that students may encounter while studying photosynthesis. However, 8 out of 10 high school biology teachers were aware of the conceptual problem of understanding [31]. Sufficient knowledge is undoubtedly important for understanding photosynthesis. A lack of knowledge states is the main problem of understanding; for example, Brown and Schwartz [19] pointed that phenomenological knowledge (understanding phenomena, the introduction of concepts) is not sufficient; it is also necessary to understand the relationships between concepts.

Understanding relationships between phenomena is harder than understanding concepts [21,36]; therefore, a lack of understanding of relationships and contexts was found in a number of analyzed articles. Anderson et al. [1] (p. 761) stated that "Most students of elementary education could not explain how animal cells use either food or oxygen. They understood plants as vaguely analogous to animals, taking in food through their roots instead of mouths." Or "Most of the students had taken at least a full year of biology, but after it, serious misconceptions persisted." Mason [37] found that Italian pupils (aged 10–11 years) knew that plants produced oxygen, but did not conclude that animals could not live without plants. Their knowledge of relationships was insufficient, as they did not realize the importance of plants for animals and for humans.

A vast number of misconceptions pertaining to photosynthesis were described in the articles analyzed. Some of them even also appeared in textbooks [38]. The two areas in which misconceptions occur most frequently have been identified:

- the relationship between photosynthesis and respiration;
- plant nutrition.

Both misconceptions were examined by Çepni et al. [39], who investigated undergraduate students in Turkey; as well as by Mason [37], who studied fifth-grade pupils in Italy.

### 3.2.1. The Misconception Connected with Plant Respiration

Cañal [5] conducted a comprehensive review of the topic of inverse respiration in plants. His main finding was that misconceptions change over the course of the education process, according to a fixed pattern. In the first years of primary school, pupils tend to believe that plants do not breathe, or that they breathe similarly to people and other animals. After learning about plant respiration, another misconception appears. In the upper primary school classes and throughout secondary school, the idea of inverse respiration appears widely.

It appears that this misconception remains persistent. Although its incidence may decrease slightly subsequently, it remains common at all levels of school and university [5] (p. 364). He pointed out that very similar misconceptions were found among students of

all ages and from countries as diverse as Australia, Spain, France, the UK, Israel, the USA, and New Zealand.

Eisen and Stavy [29] (p. 114) wrote that "the main misconception in biology was the perception of photosynthesis as a type of respiration". Specific kinds of false ideas include: "Plants respire only at night because they do photosynthesis during the day" [40], "Respiration is the reverse of photosynthesis because products of photosynthesis are reactants of respiration" [41] (p. 136).

Ng and Gunstone [42] found out that only 41% of Australian students in the 10th grade (approximately 15 years of age) answered correctly to a true–false question stating "plants carry out photosynthesis in the day and respiration in the night".

In Mamaroti and Galanopoulou [25], Greek respondents from the first class of middle secondary school (approximately 13 years of age) answered the question "When do plants respire?" As many as 27% answered that photosynthesis and respiration take place during the day, 21.3% said that photosynthesis takes place during the day and plants respire continuously, 18.5% said that photosynthesis takes place during the day, and plants respire during the night, 15.2% said that photosynthesis and respiration take place continuously, and 12.5% said that photosynthesis takes place when there is light, and plants respire continuously. Misconceptions about gas exchange were also noted by other authors, e.g., [8,19,23,31,36,43]. Švandová [44] identified the most common mistakes among lower secondary students (grades 6 to 9) from the Czech Republic: that photosynthesis is a type of respiration (for plants), and that respiration takes place only in leaves that have special organs such as pores. Mason [37] found that all pupils from Italian elementary schools (approximately 10 years of age) believed that plants breathe at night only. Flores et al. [45] found that high school students from Mexico believed that plants do not need oxygen. The few studies carried out among teachers or students in teaching disciplines produced alarming results. For example, Eisen and Stavy [29] found that 24% of the biology major and twice as many non-biology major students listed oxygen as one of the materials that plants absorb in order to build their bodies. These students had probably confused respiration and photosynthesis.

### 3.2.2. Misconceptions around Plant Nutrition

Much attention was paid to misconceptions in the field of plant nutrition in the analyzed articles. The most common problem is the misunderstanding of the differences between plants and animals (humans). For example, Barman et al. [46] conducted extensive quantitative research on 3000 pupils (grades 3–8) in the USA and often found that students had the naive idea that "plants eat and drink like humans". Respondents confused plant and animal nutrition, in that they did not grasp the idea that plants make their own food, that is, they did not recognize autotrophy [25]. Similar findings emerged from students of different ages from various countries and grades: such as Australian students aged 13–17 years [47], high school and university students from Israel [29], science teachers from the US [48], fifth-grade pupils from Italy [37], high school students from the US [13], Canadian students aged 9 to 11 years [49], fourth grade pupils from Italy [43], students from third to eighth grades in the US and high school students from Turkey [29], university students—future primary and biology teachers from Finland [31], and university students—future non-biology teachers from the US [50].

A large number of interesting data were found in the articles. There is no space to present them all. Therefore, we give only one example. Italian pupils (ages 10–11) admitted that plants are living beings as they sprout, grow, develop, reproduce, and die. Subsequently, when asked what plants need to survive, they stated the following:

- water (66.7%),
- minerals (53.3%),
- light (46.7%),
- air (33.3%),
- soil (20%),

- vitamins (6.7%),
- carbon dioxide (6.7%). [37]

### 3.3. Correcting Misconceptions

Many authors have also focused on how to correct misconceptions. All the authors of our research sample who dealt with this agreed that it is very difficult to correct entrenched misconceptions.

Österlind [51] found that while studying texts, eighth grade students from Sweden avoided conflicts with their existing views and incorporated new knowledge incorrectly into the ideas they already had. Dauer et al. [52] investigated how middle and high school students connected explanations and arguments from evidence of plant growth and metabolism. An analysis of post-instructional interviews showed that "Some students reasoned about plant growth as an action enabled by water, air, sunlight, and soil rather than a process of matter and energy transformation. These students reinterpreted the hypotheses and results of standard investigations of plant growth, to match their understanding of how plants grow." [52] (pp. 397, 408) This is how most of the students who participated in the research proceeded. Only the more advanced students consistently interpreted mass changes in plants or soil as evidence of the movement of matter.

Mason [37] stated that some of the pupils' concepts were so strong that the fifth-grade pupils in Italy quarreled over them with classmates who had different concepts. Mason pointed out the fact that the misconceptions of students are stronger than the implementers realize, hindering effective education of the topic.

Hazel and Prosser [32], in their quantitative-qualitative research among university students in Australia, deduced that it was their reluctance to abandon their pupil's concepts that caused the failure to achieve the expected success in the course. "This study revealed considerable stability in students' conceptual knowledge of photosynthesis throughout the course. There were only small improvements in the details and overall structure of students' knowledge" [32] (p. 278).

In addition, Cardac [36] cited the inability to abandon dysfunctional concepts to explain why students did not understand the relationships between concepts. Problems in cognitive learning arose when information acquired in later years conflicted with previous knowledge. From this point of view, it is better if students do not learn about photosynthesis prematurely.

Learning science in a meaningful way means realigning, reorganizing, or replacing existing conceptions to accommodate new ideas. Posner et al. [53] named the new educational approach "a process of conceptual change" [48]. Several other analyzed articles addressed the question of how to support conceptual change. For example, Çepni et al. [39] stated that computer-assisted instruction material (CAIM) specially developed for the unit of photosynthesis has the potential to correct the misconceptions of college students. They evaluated five science teachers and 52 of their students (half of them in a control group) from two high schools in Turkey. Their results illustrated that the developed CAIM influenced student's attitudes towards science lessons in a positive way. Moreover, the student's misconceptions decreased more in the experimental group. However, the expected attitude changes did not occur in the experimental group.

As many authors describe, it is a powerful experience when a student discovers and corrects their misconceptions. Modern technology can help monitor this hidden process. For example, Penttinen et al. [54] captured the process of discovering and abandoning misconceptions (specifically, oxygen was the only outcome of photosynthesis) with the help of a significant change in the eye-tracking curve. Other authors (e.g., [43,55,56]), as cited in [54], also reported similar findings.

## 4. Discussion and Conclusions

Photosynthesis is a crucial topic and provides a basis for understanding many other phenomena and relationships in ecosystems. Links with environmental issues (climate change, ozone hole, etc.) have also been discovered.

Since 1980, teaching photosynthesis has gradually come to the forefront, and the number of articles on this topic has doubled approximately every 10 years.

The teaching of photosynthesis places considerable demand on the pedagogical knowledge of teachers at all levels of education, including primary teachers. Teachers must be able to understand photosynthesis at various levels (molecular, ecological, evolutionary, etc.), which requires a high level of knowledge of physics, chemistry, and biology. They must understand the essential role of photosynthesis in the evolution of organisms and their environmental links. Finally, they must also know effective ways to teach photosynthesis, with responsibility, and to harness students' abilities.

Teaching photosynthesis can support the development of metacognition and the acquisition of scientific language, which students can then use in the study of other complex scientific topics. Therefore, learning about photosynthesis can serve us as a model for understanding the process of learning with similarly difficult and complex topics, such as the topic of evolution or the origin of man or global climate change. Photosynthesis is an appropriate topic, because it forms a part of the curriculum in all countries and because it is not controversial. The epistemological beliefs of students do not influence their understanding of photosynthesis. It is independent of religious beliefs and other cultural influences. Its understanding depends only on the quality of teaching.

**Author Contributions:** Conceptualization, K.J. and A.J.; methodology, K.J.; validation, K.J. and A.J.; formal analysis, A.J.; investigation, K.J.; resources, A.J.; data curation, A.J.; writing—original draft preparation, K.J.; writing—review and editing, K.J.; visualization, A.J. All authors have read and agreed to the published version of the manuscript.

**Funding:** This research was funded by The Ministry of Education, Youth and Sports—Institutional Support for Long-term Development of Research Organizations under Cooperatio/SOC/GEAP/ Sustainability education—Charles University, Faculty of Education (2022).

**Conflicts of Interest:** The authors declare no conflict of interest.

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
