# Peer review of "How to Teach Photosynthesis? A Review of Academic Research"

_sustainability, doi:10.3390/su142013529_

Round 1

Reviewer 1 Report

Dear authors:

I congratulate you for choosing this theme; however, reading your manuscript shows us that it is flawed. Thus, I recommend you look at my reflections/analysis (see below) on some issues.

- the title does not fully reflect the study: "Misconceptions about photosynthesis:...".

because reading (see lines 54-7) leads one to think it is "How to teach photosynthesis?" and its relation to "misconceptions in the understanding of photosynthesis".

- it seems to me that the statement is misplaced ... (lines 50-51).

- The statement "Photosynthesis is one of the most important metabolic processes on Earth" seems unscientific to me ...

- The statement "Students of biology education (future teachers in primary and secondary schools) " should be framed in the country. For example, in my country it is different.

- What is written in lines 36-37, seems to me to be a relationship that cannot be made.

- The section " 2. Materials and Methods" is very unclear. The sample selection criteria should be explained. It is unclear the period was selected for research (see what they wrote in line 63; see Fig. 1, Fig. 3...). Why PRISMA 2019 methodology and not PRISMA 2020 statement. Figure 2 should include in the legend which is the selection topic (n=80). Moreover, the graphs are of poor graphical quality. I cannot understand the interest of table 1.

-Review the statement (see lines 325-326), it is questionable, as it seems to me that it must fit into the teaching process followed.

- What do you write in lines 353-355 is a miscellany. The choice of textbooks is another problem of the teaching-learning process.

- You wrote (line 357):"(considering brain development)", but I think that as a result of a clerical error.

- I've only seen one reference in the last five years.

Rew

Author Response

- the title does not fully reflect the study: "Misconceptions about photosynthesis:...".

because reading (see lines 54-7) leads one to think it is "How to teach photosynthesis?" and its relation to "misconceptions in the understanding of photosynthesis".

Thank you for  recommendation, we changed the title of the article.

- it seems to me that the statement is misplaced ... (lines 50-51).

Thank you, we reduced and rewrote text.

- The statement "Photosynthesis is one of the most important metabolic processes on Earth" seems unscientific to me ...

We rewrote it.

- The statement "Students of biology education (future teachers in primary and secondary schools) " should be framed in the country. For example, in my country it is different.

We included word "Czech".

- What is written in lines 36-37, seems to me to be a relationship that cannot be made.

We changed the formulation.

- The section " 2. Materials and Methods" is very unclear. The sample selection criteria should be explained. It is unclear the period was selected for research (see what they wrote in line 63; see Fig. 1, Fig. 3...). Why PRISMA 2019 methodology and not PRISMA 2020 statement. Figure 2 should include in the legend which is the selection topic (n=80). Moreover, the graphs are of poor graphical quality. I cannot understand the interest of table 1.

We included information to the Figure 2 and changes figures to be in better quality.

-Review the statement (see lines 325-326), it is questionable, as it seems to me that it must fit into the teaching process followed.

We agree, but a description of the context for all citations would disproportionately increase the scope of the article. If interested, the reader can look up the context in the cited source.

- What do you write in lines 353-355 is a miscellany. The choice of textbooks is another problem of the teaching-learning process.

We agree it is another problem. We have edited the text of the article.

- You wrote (line 357):"(considering brain development)", but I think that as a result of a clerical error.

We agree, we omitted the text in brackets.

- I've only seen one reference in the last five years.

The smaller number of newer citations is related to the chosen methodology of article selection.

Reviewer 2 Report

The review comments on this manuscript are listed as follows:

1.   The literature format should follow the format of literature citation that the journal specified.

2.   The authors might ask a professional English editor to revise the manuscript.

3.   The abbreviations should have their full texts when they appear first in the manuscript.

4.   The authors should describe how they identify the articles for this study.

5.   The authors should have more explanations in Figure 1; especially, the left-up concert bar chart in Figure1.

6.   Examining Figure2, it is difficult for readers to get the results mentioned line 77 – line 80. The authors should have more descriptions related to Figure2.

7.   The number of research papers in Table 1 is 62; however, line 92 describes the numbers of the qualitative and quantitative approaches are 25 and 27. Moreover, Figure 3 shows that 72 articles were explored in this study. There exists apparent gaps between Table 1, Figure 3, and the description in line 92.

8.   The authors should explain how they get the information shown in Figure 4.

9.   About the qualitative approach, the authors described lots of articles related to photosynthesis teaching, the misconception connected with plant respiration, and correcting misconceptions; however, the authors did not have a further exploration on these topics.

10.     The authors took more space to explore the qualitative approach in the manuscript; compared with the qualitative approach, the authors should have further explorations about the quantitative approach.

Author Response

  1.   The literature format should follow the format of literature citation that the journal specified.

We checked formatting again.

  1.   The authors might ask a professional English editor to revise the manuscript.

The text was checked by a native speaker.

  1.   The abbreviations should have their full texts when they appear first in the manuscript.

    Was added.

    4.   The authors should describe how they identify the articles for this study.

A more detailed description has been inserted into the text.

  1.   The authors should have more explanations in Figure 1; especially, the left-up concert bar chart in Figure1.

Added information that there is a cumulative graph in the upper left total.

  1.   Examining Figure2, it is difficult for readers to get the results mentioned line 77 – line 80. The authors should have more descriptions related to Figure2.

Information inserted in image description.

  1.   The number of research papers in Table 1 is 62; however, line 92 describes the numbers of the qualitative and quantitative approaches are 25 and 27. Moreover, Figure 3 shows that 72 articles were explored in this study. There exists apparent gaps between Table 1, Figure 3, and the description in line 92.

The numbers are correct. Table 1 uses journals where only 5 and more articles on the chosen topic were published. The description on line 92 and beyond has been modified for better understanding

  1.   The authors should explain how they get the information shown in Figure 4.

Data were obtained by analysing the articles by the authors.

  1.     The authors took more space to explore the qualitative approach in the manuscript; compared with the qualitative approach, the authors should have further explorations about the quantitative approach.

Thanks for the recommendation. We are also preparing to process the second part, which concerns quantitative approaches.

Reviewer 3 Report

The authors present an interesting study whose objective is to conduct a critical review of the subject of photosynthesis education by examining 80 systematically selected articles using quantitative and qualitative approaches. The manuscript is written in a very clear and interesting way. The study method is described in an appropriate way. The summary and conclusion are supported by the content. The work presented remains original since, in terms of plagiarism assessment, only about 20 % of this document consists of text more or less similar to the content considered most relevant by iThenticate. I believe this is an interesting contribution and I recommend it for publication in Sustainability. 

Author Response

Thank you for your kind review.

Round 2

Reviewer 1 Report

Dear authors:

This  sentence "Following the procedures established in the PRISMA statement (...)" should be precise. Thus, I suggest: "Following the procedures established in the PRISMA 2019 statement (...)"

Rew

Reviewer 2 Report

The authors depended on all comment to revise the manuscript.